# Going to the Morgue with Andres Serrano: Provocation as Revelation

Alex Sosler

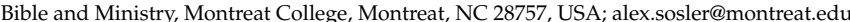

Bible and Ministry, Montreat College, Montreat, NC 28757, USA; alex.sosler@montreat.edu

**Abstract:** Originally displayed in Paula Cooper Gallery in New York City's SoHo district, Andres Serrano's *The Morgue* series continued the artist's controversial and transgressive work. Set against a black backdrop in a mortuary, he photographed dead bodies in different stages of decomposition. In this article, I borrow from Charles Taylor's cultural analysis of the secular and Flannery O'Connor's literary theory of the revelatory power of the grotesque to discuss Serrano's artistic choices. In essence, I argue that his work is not a desecration of humanity but a stark reminder of the sacralization of humanity. As such, Serrano's work is not provocative for provocation's sake, but a provocation to poke holes in a disenchanted age. Underneath Serrano's images is the question: if this is a heap of flesh, why are you provoked? In a culture that avoids death at all costs, Serrano reminds the contemporary world of their mortality with an updated form of *memento mori* art.

**Keywords:** Andres Serrano; transgressive art; death and dying; mortality; secularization; Flannery O'Connor; grotesque





## 1. Introduction: A Secular Age and the Avoidance of Death

Andres Serrano (b. 1950) was raised Catholic and regularly uses religious symbols as a sort of iconography. Even as he rejects the church's dogma, he is drawn to the church's aesthetic, which makes him an interesting subject of theological exploration. Though trained as a sculptor and painter in high school, like a conceptual artist, he entered the art scene of photography with no formal training. In this article, I examine Serrano's *The Morgue* (Serrano 1992) as it provokes the modern viewer who has forgotten death. In many ways, his work is a contemporary memento mori series. Rather than skulls, withering flowers, and ticking clocks, Serrano confronts the modern viewer with actual corpses in a morgue to remind them that they will die. I will draw on the work of both Charles Taylor and Arthur McGill to describe how modern people tend to avoid death. Considering the avoidance of death, Serrano's provocative images reveal a neglected truth about death similar to Flannery O'Connor's literary theory of the grotesque as revelatory.

Charles Taylor's tome entitled *A Secular Age* attempts to describe the "feel" of the modern age. It is less a purely philosophical treatise about what ideas have shaped modernity and more of the modern story of what is believable. As he artfully argues, the secular age is not an age of non-belief or the opposite of belief but an age of competing beliefs. Believing in the transcendent is more difficult than it was 500 years ago—not because we have progressed scientifically to explain away all mystery—but because we have changed the plausibility of belief. The way modern persons measure what is believable has changed. At its root, the secular age is not a change to "non-belief" but a change of belief. As such, the narrative of a secular age is different from the medieval story.

In the medieval imaginary, the self was porous; that is, the human person was open to the spiritual, transcendence, mystery, and the divine. God could and did interact with the world. However, at the heart of a secular age is the idea of a buffered self (Taylor 2007, pp. 38–39). Rather than being open to the transcendent, modern humanity has cut itself off from the divine (thus, the idea of a buffer between the self and the divine).

Persons exist only in the temporal and chronological mode. In other words, the meaning, purpose, and significance of one's life must be found in between the time of birth and the time of death, because that is all one has. The self is insulated and isolated in its individuality. There is no inheritance to steward, and there is nothing after death. In biblical parlance, it is similar to the passage in Ecclesiastes where the Preacher is trying to find meaning and purpose in life "under the sun". The continual refrain becomes "vanity of vanities".

In approaching death in a secular age, ignorance seems to be the best option one has. Taylor writes:

> "We don't know how to deal with death, and so we ignore it as much and for as long as possible. We concentrate on life. The dying don't want to impose their plight on the people they love, even though they may be eager, even aching to talk about what it means to them now that they face it. Doctors and others fail to pick up on this desire, because they project their own reluctance to deal with death onto the patient. Sometimes the dying will ask that their loved ones make no fuss over them, hold no ceremony, just cremate them and move on; as though they were doing the bereaved a favour in colluding in their aversion to death. The aim can be to glide through the whole affair, smoothly and as much as possible painlessly, for both dying and bereaved." (Taylor 2007, p. 723)

Whereas the medieval imaginary saw suffering as inevitable and something to enrich and strengthen the self, the secular imaginary sees no purpose or meaning in suffering at all. And central to death is suffering. Life is for happiness and health—for living. Suffering destroys happiness—and death is the final defeater. So rather than consider death and its looming reality, modern persons tend to ignore the final suffering, having neglected their temporal suffering.

Churches are often not much better than their alternatively believing counterparts. Religion has adopted the avoidance of death as the ethic of mortality. Rather than funerals, churches host "celebrations of life". Rather than preparing people for death, churches provide a theology lived within the "immanent frame" (Taylor 2007, pp. 542–50). As this is the only life to live, flourishing and happiness must be found in the here and now. Even the theological self seems buffered within the immanent frame: enclosed and protected from the transcendent and divine, seeking meaning and purpose in this life alone. Suffering is a mere interruption to the best life now.

Arthur McGill calls this approach to death the ethic of avoidance. Americans, according to McGill, are after success. Pain, suffering, and neediness are things to be avoided at all costs. In modern language, when greeted with the popular "how are you?", even if the answer is "terrible", we reply "fine!". We would rather others be under the illusion that we are all right and not in need. McGill notes, "The whole American venture into health and success and beauty and vigor, the whole existence which Americans try to live with all their consciousness rests on the hidden substratum of suffering and dreadful knowledge" (McGill 1987, p. 37). Yet death comes as the ultimate moment of need—and there is nothing that says failure and need like death. So, McGill argues, Americans go on living their "successful" life and are never prepared to die.

In the secular age and as a buffered self, the meaning of life is not related to death at all. The only meaning one can have in life is extending one's life. As Arthur McGill contends:

> "Americans like to appear as if they give death hardly any thought at all. Of course, death will happen to all of us someday, but until then, it is not something to think about or grapple with. Until then our preoccupation should be with life and with all its challenges and adventures. Let the dying deal with death. Our calling is to enjoy life. In this regard the popular view and what I have just called the medical view of death are essentially alike . . . Both think of death as something future, normally as something remote and hypothetical—not as something that hangs over and works within every present moment of life." (McGill 2013, p. 13)

This medicalized view of death sequesters the reality of death out of view. As a matter of statistics, death has become sterilized behind the closed doors of hospitals. In the 1940s, most people died in their homes in the presence of loved ones and family. In the 1980s, 17 percent of people died in their homes (Gawande 2017, p. 6). Infant mortality rates were much higher. Graveyards were near city centers.

But today, it is easier to avoid death than in previous generations. Death, a fate inevitable in previous generations, is seen as unnatural to life. Death is something not to reckon with but to push further out of the bounds of one's mind. On this point, Arthur McGill (2013) is incisive: "This means that death is outside of all value and all meaning, because for Americans value and meaning belong exclusively to life. For anyone to accept death or dignify death would be to deny human value and dignity. Death is the total enemy" (McGill 1987, p. 17). Those living within a secular age are always pursuing some future experience, as long as the future never includes death.

## 2. The Conceptual Context for Serrano's Work

Andres Serrano is likely not the first name that comes to mind when one thinks of conceptual art. Transgressive and unconventional, sure—but I want to contend for Serrano as a type of conceptual artist. Typically, conceptual art emphasizes an idea over a style, as argued by Anthony Janson (" . . . with conceptual artists, idea, concept, or information will be the consuming quality of the work" (Davies et al. 2010, p. 1062)) or Elizabeth Schellekens (" . . . for most conceptual artists, artistic value is only to be gained from the knowledge, insight, or understanding that artworks may generate" (Schellekens 2005, p. 80)). I link Serrano to conceptual art because, as Schellekens writes, Serrano is generative.

To be clear, it is not that Serrano is absent of style or quality, but that the power of his photographs lies beyond the image, in the ideas behind them. This understanding concurs with both Alberro and Stinson on conceptual art in general and Hopkins on conceptual art in particular. Alberro and Stinson argue that " . . . Conceptual Art did not reduce (or attempt to reduce) the pictorial to the linguistic (or textual). The point is, rather, that the gaps and connections, the lemmas and absurdities between the pictorial and the textual, are spaces in which much cultural aggravation was and is possible" (Alberro and Stimson 1999, p. 445), while Hopkins suggests that " . . . photography was primarily a social sign system whose operations needed to be 'exposed'" (Hopkins 2018, p. 180). I am suggesting that these gaps and connections between the visual and philosophical are evident in Serrano's photographs. His work is suggestive, with layers of meaning that generate creative connections. As such, the aesthetic ingenuity is less important in Serrano's work, and the generative connections lie behind and beyond the subject of his photography. He is not generating a new stage in photography as much as he is focusing on the conceptual through his choice of content. This is the nature of his provocative work: one does not have to appreciate it or model art after him, but his work does reveal and expose on multiple levels. Or, as art critic Michael Brenson wrote on Serrano's work in the *New York Times*, "It is the photograph that becomes the vessel of transformation and revelation" (Brenson 1989).

*The Morgue*, a series of photographs originally appearing in New York City's Paula Cooper Gallery in 1992, is a direct affront to the avoidance of death in today's world. The photographs, typical of Serrano, are large prints—in a way, making the images unavoidable. Serrano's subjects were photographed at an unnamed morgue against a black backdrop. There are two types of morgue: those that house the corpses of persons who have passed away with a known cause of death (medical), and those where the cause of death is unknown (criminal). Based on the types of photograph that Serrano takes and the names he gives to his pieces, it is safe to assume that the morgue he used was a criminal morgue, where the cause of death was either uncertain or related to violence. This reality presents an added nuance to interpretating Serrano's work: there is a criminal atmosphere as Serrano takes the viewer to the morgue. The bodies are unidentified, and in order to obtain permission for the photographs, Serrano had to portray them as unidentifiable. The subjects are in different stages of decomposition, having died from different causes—AIDS,

rat poison, stab wounds, burning. As such, none of the victims passed peacefully in their sleep, and the viewer is often confronted with a premature death.

Regarding his own work, Serrano reflects that, "I see myself as belonging to a tradition of religious art going back to Caravaggio and others . . . Caravaggio's works are so strong—using a prostitute as the Virgin Mary . . . " (Jones 2016). This religious aesthetic combined with the provocative image has come to define Serrano's work, and the same runs true in *The Morgue.*

In an interview with *Bomb Magazine*, the interviewer Anna Blume tries to pin down what exactly Serrano is trying to do with his series. She asks, "Your images draw us in and they are beautiful at a certain level. You bring us close, very close to details of dead bodies, which sets off an alarm of feeling and thinking, but all this stops on the surface and we are left as voyeurs rather than as witnesses of death. I wonder as I look at the photographs what are you trying to do with this show, what are you doing with dead bodies?" Blume highlights another aspect of the photographs—not only are they large, but many of them are shot up close. To which Serrano responds, "The morgue is a secret temple where few people are allowed. Paradoxically, we will all be let in one day. I think you're upset and confused that I've brought you there prematurely. My intention is only to take you to this sacred place. The rest is entirely up to you. I explored this territory with fresh eyes and an open mind. I want the audience to do the same and to see it's a process of discovery for me too." (Blume 1993). Serrano will not tell the viewer what he is trying to do or comfort the interviewer with pleasant thoughts about the morgue. Everyone must encounter the images individually, because that is the way each person encounters death: alone. Serrano invites the viewer into the process of discovery about their own mortality.

Later in the same interview, growing more perturbed, he says, "Because you want me to lead you by the hand and make you feel like these are images that you can see in a political or social or feminist way that would fit with your thinking and if I don't tell you you're right or point you in that direction, then you feel abandoned." (Blume 1993). This sense of abandonment in the face of death is common in our secular age. To pin down exactly what Serrano is attempting to portray is something that Serrano himself will not allow. These different ways to interpret Serrano led to criticism from multiple perspectives.

The reception of Serrano's work has been, of course, controversial. His work can be evaluated through multiple lenses because he will not "lead you by the hand" (Blume 1993) to the correct interpretation, so it has and can be critiqued through multiple readings. On the theological side, Carl Trueman names Serrano in his recent work entitled *The Rise and Triumph of the Modern Self.* He refers to artists like Serrano as "deathworks", borrowing the phrase from Philip Rieff (Trueman 2020, pp. 96–100). In essence, a "deathwork" is an "attack on established cultural art forms in a manner designed to undo the deeper moral structure of society" (Trueman 2020, p. 96). In Trueman's reading, Serrano represents a tearing down of the moral framework regarding the sacredness of the human and the dignity of death, thus uprooting Judeo-Christian values. By so doing, Serrano desecrates what traditional society holds dear and the basis of moral foundations. Referring to Serrano's *Piss Christ*, Trueman writes, "Serrano is not simply mocking the sacred order in this work of art; he has turned it into something dirty, disgusting, and vile . . . The sacramental is made into the excremental." (Trueman 2020, p. 97). Likewise, with the nudity contained in *The Morgue*, one could argue that he turns the dignity of death into something pornographic. But I think Serrano himself would be surprised by both readings.

In contrast to the theological, Serrano is also disparaged from a feminist perspective. Referring to *Rat Poison Suicide, III*, wherein a female corpse is portrayed with incisions running from her exposed vagina up through her chest, Andrea Fitzpatrick writes that "Serrano's photographic approach exposes her in a disturbing way—by prodding her and prying her open for view." (Fitzpatrick 2008, p. 37). In Fitzpatrick's viewpoint, this exposé is without consent or agency and thus degrades the woman in view. As such, these representational images are "disrespectful, if not also wounding" (Fitzpatrick 2008, p. 28).

To be fair, both the theological and feminist readings of the images are not without merit: there is something provocative and potentially degrading in viewing the dead or submerging a crucifix in urine. Viewing dead, and sometimes naked, bodies is not something that seems right or ordinary. It is certainly uncomfortable. However, I want to present an interpretation that Serrano's work does not tear down moral foundations or prey on the vulnerability of the dead but reveals deeper truths through shock.

## 3. Revelation through Provocation

In a letter to a friend, Flannery O'Connor wrote, "The truth does not change according to our ability to stomach it emotionally. A higher paradox confounds the emotion as well as reason and there are long periods in the lives of all of us, when the truth as revealed by faith is hideous, emotionally disturbing, downright repulsive. Witness the dark night of the soul in individual saints." (O'Connor 1988, p. 99). When one encounters O'Connor's fiction, one can understand the above quote. She leans into the repulsive and disturbing and does not shy away from it. She presents truth without concern for how the reader feels about it or their ability to stomach it. In writing fiction, she portrays reality as it is—whether one is repulsed by the real or not. Serrano would find a kindred spirit in O'Connor. In more fully describing her literary theory, O'Connor is worth quoting at length:

> "In these grotesque works, we find that the writer has made alive some experience which we are not accustomed to observe every day, or which the ordinary man may never experience in his ordinary life ... Yet the characters have an inner coherence, if not always a coherence to their social framework. Their fictional qualities lean away from typical social patterns, toward mystery and the unexpected. It is this kind of realism that I want to consider. All novelists are fundamentally seekers and describers of the real, but the realism of each novelist will depend on his view of the ultimate reaches of reality." (O'Connor 1961, p. 40)

In another place, she writes, "The novelist with Christian concerns will find in modern life distortions which are repugnant to him, and his problem will be to make these seem like distortions to an audience which is used to seeing them as natural; and he may well be forced to take ever more violent means to get his vision across to a hostile audience ... to the hard of hearing you shout, and for the almost-blind you draw large and startling figures." (O'Connor 1961, pp. 85–86). What O'Connor accomplishes in her fiction, I propose that Serrano achieves in his photography. In a world that is deaf in regards to death, Serrano is shouting about the natural end of human life. He shows the real, and the real does not depend on a culture's ability to stomach it. Often, the real leans toward the unexpected and mystery. It is not everyday that we witness the stark reality of death—even violent death—but Serrano leads us there to discover something. He uses the grotesque to make us alive to some experience to which we are no longer accustomed.

## 4. Serrano's Memento Mori

To return to Taylor's *A Secular Age*, the author describes the resulting "feel" of the nova effect as malaise. By this, he means the general discomfort one feels having so many options and being pushed and pulled by enchantment, disenchantment, transcendence, and immanence. I propose that amid this malaise, one's condition needs to be shocked in order to see the real. While death is rationalized away as something unnatural and to be avoided or it is claimed that the deceased are in a better place now, Serrano confronts us with those truths. In a closed and immanent frame, these bodies should mean no more to the viewer than roadkill. The bodies viewed are mere flesh and bone in the modern sense. Yet this view seems disrespectful and uncomfortable. Even as the viewer does not know who these bodies belonged to in life, there seems to be something wrong about viewing them in their vulnerability. This shock gives way to mystery and the unexplainable—the very thing the secular tries to explain away and limit through avoidance. In the modern age, everything should be explainable. The grotesque is like the defibrillator that shocks one's heart to feel the reality the secular age may have forgotten. In the morgue specifically,

the reality of death provokes the viewer to remember that humans are more than molecules. It may seem that Serrano drags the viewer to the morgue as a voyeur, but my contention is that he takes us there as witnesses to the real.

Whereas past memento mori art featured subtle images of ticking clocks and decaying plants to remind their viewer of death, Serrano's images are needed to provoke the modern consciousness. Perhaps those in the past were acquainted with death and grief, so all they needed was the subtle image. Contemporary people are less acquainted with the reality of death and so find it easier to avoid. In such a cultural situation, Serrano creates a conceptual link rather than a stylistic link to the memento mori visual tradition: skulls with flesh still on them remind us of death better than hourglasses or withering flowers. These grotesque images shock the modern viewer into remembering their death. Traditional memento mori representations used subtle images to remind the viewer of the stark reality; Serrano uses the stark reality to reveal a subtle theology. In the following section, I will highlight the subtle theology of Serrano's grotesque photographs. In essence, he displays the sacredness of life and the holiness of death, creaturehood, and vulnerability, and the power of contemplation and attention.

### 4.1. Sacred Life and Holy Death

While the secular suppresses the spiritual or transcendent, Serrano's images portray a life or soul that is unavoidable. As much as these images picture death, they also display a hidden life. In other words, if these images portray mere biological components and fleshly atoms combined to form decaying matter, why is the viewer provoked and bothered? Serrano concludes on his own work: "Well, I won't say that I believe in a soul. But I do believe that I've captured an essence, a humanity in these people. For me, these are not mere corpses. They are not inanimate, lifeless objects. There is a sense of life, a spirituality that I get from them. This is an important point for me. There is life after death, in a way." (Blume 1993). These images continue to give life even after death. There is a sense of the soul—and the viewer is confronted with these truths even if they deny them.

These reflections by Serrano on the soul and life after death again echo theologian Arthur McGill:

> "An obituary tells of what this person has received by the expenditure of others, above all by that of his parents and family, but also by that which came from all the anonymous ones who sustained him. At the same time an obituary also records this person's life, that is, the course of his self-expenditure, *in the context of his dying* . . . In this Christic experience of the 'newly dead', we do not have a theophany of the absoluteness of death. We have the integration of death with the life of giving and receiving." (McGill 2013, p. 73)

Serrano creates these images as a sort of obituary. Death has not ended the corpse's life, but their life extends through these images in the same way as life continues through the memory of the dead. In actuality, the avoidance of death cheapens one's life, and these memento mori pictures are an opportunity for the viewer to deepen one's own life through the life of another.

The most regular critique of Serrano is that he provokes for provocation's sake—that he desecrates some moral framework. One could argue that Serrano is not provoking for no purpose; rather, he shocks the viewer to see the truth. Perhaps Serrano is not desecrating death but resacralizing death. He does not take the viewer to the morgue to trivialize or cheapen death but to enrich it. And to consider the holiness of death makes the modern person uncomfortable, as they have sought to avoid the truth about the end of themselves.

As David Deitcher has written, "Serrano's photographs are therefore indefensible to so many, not simply because they blaspheme, which, in a narrow sense, some of them do; they are insupportable because they find beauty in substances that have always made people recoil in horror and embarrassment." (Deitcher 1989, p. 141). The recoiling is part of the revelation. Like O'Connor, Serrano shocks the viewer so that he or she can see more

clearly. In so doing, he does not desecrate dead bodies but reveals the sacralization of life in the presence of death. To see that sacredness, our apathy needs to be disturbed.

### 4.2. Creaturehood and Vulnerability

A central distinction in Christian theology is the Creator–creature distinction. At the beginning of Genesis, God speaks into a formless and void space and begins to form the world (days 1–3) and fill (days 4–6) the emptiness. The climax of this week is the creation of man and woman. The neediness of creaturehood is explicated in the rest of the Bible and Christian tradition. All creatures needed God for the beginning of life, and they continue to need throughout life. The heart of idolatry is replacing the Creator God with some sort of creature (Rom 1:22–23) and elevating creation above the Creator. So as sacred as life and death are, they both stand in dependence on God, who "gives and takes away" (Job 1:21).

To return to the idea of the grotesque found in O'Connor, both O'Connor and Serrano reveal truths contextually. O'Connor wrote in the Christ-haunted South. People thought they were generally good and morally virtuous. Yet her fiction comes with an emotional shock: "No, you are not 'good country people.'" In comparison, Serrano's photographs first appear in a secular New York City, where the common conception is that humans are material people part of a material world. Living and breathing is all we have, and then death occurs, and then nothingness. Serrano presents his images to show and shock: "No, you are more than material. You live in need and dependence and avoiding that fact does not make it go away".

So, Serrano is presenting us with images reminding us of our own impending death and our status of need. Whereas most modern Americans ignore death, these photographs make the viewer consider death and creaturehood. Jesus comes with "dying as the focus and center of his existence" (McGill 1987, p. 46). From early in the gospel narratives, his face was set toward Jerusalem to die. He lived in constant vulnerability, considering his own impending death, and depended on the Father to tell him what to do and say. He did nothing on his own accord (John 5:19; 12:49).

Need creates a posture toward life that is bent toward vulnerability. Need means that Christians are constant receivers of identity rather than curators of identity. The way the Christian lives is by constantly receiving and constantly expending. McGill calls this "receiving without having" (McGill 1987, p. 61). Taking up the cross of Christ is a move to need and dependence; it is a very real death which creates very real need. Life is not about self-actualization or self-fulfillment, but considering the life of Christ, life is about self-expenditure. As such, "Jesus Christ establishes a celebration of neediness because neediness is indispensable for receiving—and receiving is the saying yes to love" (McGill 1987, p. 86). To be broken open into need is to be open to receive something one does not have. The only way that persons can be ready to receive love is to recognize their need of it. Therefore, the realization of need prepares us for the ultimate expression of need in death. Becoming familiar with death allows a person to live well.

For the Christian, then, vulnerability is the normal mode of existence. The ultimate vulnerability is death—something we cannot control or plan. Arthur McGill argues as much when he writes, "Death simply shows the essential emptiness of the creature wherein he depends upon God for all his being. Death discloses the extent of having—its total vacuity. Death is therefore that neediness in which the creature always stands to his creator, in which the children of God always stand in relation to their Father. Death is the disclosure of yawning need . . . " (McGill 2013, p. 44). Christians are those who have become acquainted with death because they are acquainted with need. Death is the ultimate vulnerability of human needs.

Serrano, then, is presenting a theological image. He is reminding the viewer of need—a need they cannot escape or ignore. Death will meet each person, and it requires a reckoning. Perhaps the thing that the modern viewer is most uncomfortable with is a reminder of their own vulnerability.

*4.3. Contemplation and Attention*

Lastly, key to Serrano's art is the attentional nature of the theological and artistic task. The modern age is marked by a desire to scientifically know and explain. This effects even the Christian task of knowing. The desire is for a theology we can master and explain rather than behold and contemplate. But all theology starts in silence and ends with wonder. The monastic traditions of Christianity preserve the countercultural witness of this contemplative spirit. True art, too, cannot be condensed into a philosophical syllogism stating its meaning. As such, the artistic task and the theological task share an attentive posture and contemplative orientation.

In the interview with *Bomb Magazine*, Serrano states, regarding his images, that "They engage the viewer in a dialogue that is difficult to escape. You can turn the pages of a newspaper or flip a channel easier than you can walk out of a gallery. The curious thing is that most people who have seen this work are compelled to stay." (Blume 1993). This idea has resonance with another thought from Flannery O'Connor. In *Mystery and Manners*, she writes, "Any discipline can help your writing . . . Anything that helps you see, anything that makes you look. The writer should never be ashamed of staring. There is nothing that doesn't require his attention." (O'Connor 1961, p. 84). For the writer so with the photographer: there is nothing that does not require his or her attention. In another place, O'Connor adds, "Fiction is about everything human and we are made out of dust, and if you scorn getting yourself dusty, then you shouldn't write fiction. It's not a grand enough job for you." (O'Connor 1961, p. 68). Serrano is picturing the literal dust in which humans were made and to which they will return. He is not afraid or hesitant to portray the most human realities. Even in the most human realities—urine, blood, death, etc.—beauty can be seen; this contemplativeness requires attention. I suppose it would be easy to view Serrano's images and discard them as offensive and morally vacuous—not worth paying attention to. But good art demands attention, and perhaps we have not sat with the images long enough to see the beauty that resides even in death.

For the Christian, contemplating dead bodies is a weekly habit. Each time a Christian walks into a church, they see a corpus, a dead body hanging from the cross. Quite literally, death is the center of a Christian's worship aesthetic. And if death is central, need is normal. To return to the scheme of conceptual art, the power of Serrano's art lies in its aesthetic appeal, which demands attention. His photographs attract attention in a way, as Serrano says, that the viewer cannot escape. A reader can put down a book about death, but there is something valuable about the aesthetic image that forces the viewer to reckon with the reality of death and the subtle theology within the photographs. There are questions that seeing a dead body summons—especially given the detailed focus and large scale of the images.

## 5. Conclusions

As described, Serrano will not tell the viewer what to feel or how to interpret his images of dead bodies. But he is presenting a true image: "for you are dust, and to dust you shall return" (Gen 3:19). To be a creature is to be limited, fragile, and needy. To be a human is to be mortal, which means reckoning with death. In light of death, the desire for eternity—"a desire to gather together the scattered moments of meaning into some kind of whole" (Taylor 2007, p. 720)—is not some childish dream but a human desire. Death can provide a meaningful and comprehensible whole. To the modern eye, death comes as a fissure to the buffered self.

Serrano invites us to see the fracture in a purely secular world—but it requires some provoking that only the grotesque can accomplish. In the modern imagination, the body presented is a hunk of flesh. Is the rotting and decomposition of our bodies the only future? Or is there a sacredness to the human body beyond the material? Why is there a sense of over-intimacy—that you should not be seeing this? Serrano presents a nagging reminder to a broadly secular audience: remember you will die. And in light of this common reality of death, perhaps there is more to life than material or scientific existence.

By so doing, Serrano presents the plausibility of some sort of faith. In previous ages, the cultural imagination assumed many theological claims: living with death in mind, life after death, creaturely dependence, etc. These assumptions are typically not modern assumptions. Serrano does not present a full and robust theology, but his photographs draw the attention of a secular audience to reconsider their own assumptions. The images will not lead to a specific faith tradition, though they may captivate viewers to question their own plausibility structures. A secular audience may not be argued into seeing the holiness of life and death or the neediness inherent in being a creature; but Serrano makes his audience see it and feel it. There is a visceral demand that his images evoke. They unsettle. But in unsettling, fissures begin to emerge in a secular age that has buffered itself from a sense of transcendence.

**Funding:** This research received no external funding.

**Institutional Review Board Statement:** Not applicable.

**Informed Consent Statement:** Not applicable.

**Conflicts of Interest:** The author declares no conflict of interest.

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
