# Peer review of "Going to the Morgue with Andres Serrano: Provocation as Revelation"

_religions, doi:10.3390/rel13060562_

Round 1
Reviewer 1 Report
Excellent article and original perspective. Glad you took Serrano as a case study for a new way of making theology.
Author Response
Thank you for your encouragement and support!
Reviewer 2 Report
I really appreciated the opportunity to review this paper and I think that it engages with some critical ideas that make an excellent contribution to scholarship around this issue. Reading the article did make me want to see the images myself and so if it is at all possible it would be good to include an image of the work under consideration in the paper.
I do have one recommendation for the author's consideration. On Page 2 or 9, line 71 the author writes"....there is nothing after death. In biblical parlance, it is life in Ecclesiastes: life "under the sun" with no existence or help beyond human rationality." I think that this is an inaccurate representation of the message in Ecclesiastes. Whilst Ecclesiastes does focus our attention on the limitations of human existence and experience, it also reminds us that"... whatever God does endures forever; nothing can be added to it, nor anything taken from it; God has done this, so that all should stand in awe before him." (Ecc 3:14). It is not helpful or necessary to use BE in the section on Secular Age and the Avoidance of Death, and I would recommend removing this reference.
Author Response
Thank you for your reflections and concerns.
In the updated version, I have removed that reference to Ecclesiastes, as I think you are right, and I was at least unclear on what I meant. I updated it to say, "In biblical parlance, it is life in the beginning of Ecclesiastes where the Preacher is trying to find meaning and purpose in life “under the sun.”" I was trying to get to this idea at the beginning of Ecclesiastes where the Preacher is pursuing riches, wisdom, knowledge, etc., and finds them meaningless or vanity. To your point, it's in considering God "above the sun" (if I can have the freedom to use that phrase) that I think Ecclesiastes is getting at. I'm not committed to that, and I don't think it makes a big difference to the overall paper, so if you still think it's unhelpful, I'll remove it.
I have also added a hyperlink at the beginning of the paper to images on Serrano's website to give the reader a taste of the images discussed.
Reviewer 3 Report
This is an interesting article. I am only going to suggest a few changes:
Line 61: change "no belief" by "non-belief".
Line 69: this is better: ...be found in the time of birth to death".
Line 125: change "lfie" by "life".
Line 189: this is better: ...one could argue that he turns...
Line 206: better: In a letter to a friend... (without initial A)
Author Response
Thanks for this helpful feedback and grammatical eye. I really appreciate it! All your suggestions have been modified.